# Rivers’ Water Level Assessment Using UAV Photogrammetry and RANSAC Method and the Analysis of Sensitivity to Uncertainty Sources

**DOI:** 10.3390/s22145319

**Published:** 2022-07-16

**Authors:** Nicola Giulietti, Gloria Allevi, Paolo Castellini, Alberto Garinei, Milena Martarelli

**Affiliations:** 1Department of Mechanical Engineering, Politecnico di Milano, 20156 Milan, Italy; nicola.giulietti@polimi.it; 2Department of Industrial Engineering and Mathematical Sciences, Polytechnic University of Marche, 60131 Ancona, Italy; p.castellini@staff.univpm.it; 3Department of Engineering Sciences, Guglielmo Marconi University, 00193 Rome, Italy; a.garinei@unimarconi.it; 4Idea-Re S.r.l., 06128 Perugia, Italy

**Keywords:** water level, river, photogrammetry, point cloud, plane extraction, RANSAC

## Abstract

Water-level monitoring systems are fundamental for flood warnings, disaster risk assessment and the periodical analysis of the state of reservoirs. Many advantages can be obtained by performing such investigations without the need for field measurements. In this paper, a specific method for the evaluation of the water level was developed using photogrammetry that is derived from images that were recorded by unmanned aerial vehicles (UAVs). A dense point cloud was retrieved and the plane that better fits the river water surface was found by the use of the random sample consensus (RANSAC) method. A reference point of a known altitude within the image was then exploited in order to compute the distance between it and the fitted plane, in order to monitor the altitude of the free surface of the river. This paper further aims to perform a critical analysis of the sensitivity of these photogrammetric techniques for river water level determination, starting from the effects that are highlighted by the state of the art, such as random noise that is related to the image data quality, reflections and process parameters. In this work, the influences of the plane depth and number of iterations have been investigated, showing that in correspondence to the optimal plane depth (0.5 m) the error is not affected by the number of iterations.

## 1. Introduction

During the past decades, several efforts were made in monitoring geophysical and environmental parameters by the use of unmanned aerial vehicles (UAVs) [1,2,3]. In particular, this technology is widely used for checking river and lake conditions [4,5,6] in terms of their bathymetry and water level estimation. In fact, ground-based observations are not always feasible because of the high purchase and maintenance costs that are associated with gauging stations’ devices [7]; furthermore, space missions face several limitations in terms of both their spatial and temporal resolutions, making these observations not possible for this kind of investigations [8]. Bandini et al. [8] and Kasvi et al. [9] have provided an assessment of the state of the art of the principal UAV solutions, i.e., radar and sonar systems, camera-based laser distance sensor, echo sounding systems and aerial imagery. In this article, the focus is put on the latter method; more precisely, the focus is on the application of photogrammetry for the determination of rivers’ water levels. To this end, an assessment of the state of the art reveals that this technology can be implemented both for bathymetry, i.e., for rivers’ depth determination, and for water level (height above mean sea level) evaluation. The first aspect, bathymetry, is of great importance for mapping channel depths, channel morphology and the dynamics of fluvial systems [10,11]. In the pursuit of this, an approach that is called structure from motion (SfM) is diffusely exploited as it is suitable for application to images that have been taken from low-cost cameras [9,11,12,13,14]. This method aims at producing a surface reconstruction, starting from a dense point cloud, and then at finding a correlation between the river’s depth and an image-derived parameter X that is related to optical band ratios. More precisely, this parameter can be evaluated as the ratio between either the radiances or raw digital numbers of the bands that are centered in two different wavelengths [15]. Therefore, here the issue consists of selecting the optimal wavelength combination that gives the best correlation with the river’s depth. Alternatively, the same parameter can be determined by using a linear transform, as described in Legleiter [10], wherein X is calculated by means of the natural logarithm of the difference between the minimum digital number within the channel and the digital number of each in-stream pixel. In both cases, X can be considered as a parameter that is related to RGB values. This methodology in which X is evaluated as a band ratio is called optimal band ratio analysis (OBRA). Hence, by subtracting the so-derived river’s depth from the previously reconstructed water surface, one is able to determine the river bed’s elevation. In Legleiter [15] a special toolkit that is based on these concepts has been developed, integrating these latter methods with preliminary image analysis tools in order to extract a water-only image and to reduce spurious points. Niroumand-Jadidi et al. [16] presented an optimization of the OBRA technique that was aimed at overcoming the problems that are related to the choice of the optimal band ratio. This approach is called the sample-specific multiple and ratio technique (SMART) and it divides the feature space of the spectral data before creating, for each subspace, a different band ratio model that is the optimal one within that subspace. Despite a good level of accuracy, these techniques imply the need for field measurements in order to perform the correlation between depths and the RGB value [11,12,15]. Legleiter et al. [10] developed a specific algorithm in order to overcome this limitation. This algorithm links the linear relationship between the river’s depth and the X parameter to basic equations of open channel flows: continuity and flow resistance. However, despite having developed a strategy that is able to avoid field measurements, some input data are still required: an approximated river aspect ratio and flow discharge, the minimum expected depth and the water surface slope. On the other hand, Mandlburger et al. [17] were able to provide the needed reference/input data for extracting a depth estimation by photogrammetric methods by the use of a neural network called “Bathynet” that has been developed by combining photogrammetric and radiometric methods. In particular, 3D water surface and water bottom models that are derived from simultaneously captured laser bathymetry point clouds serve as the reference and training data for both the image preprocessing and actual depth estimation.

On the contrary, regarding the determination of the water level (the height above mean sea level), approaches are slightly different. In fact, the evaluation of rivers’ depths is not of interest, thus algorithms like OBRA and SMART are not required. Real-time water-level monitoring systems are particularly important for flood warnings and disaster risk assessment [18,19]. Even if the works that have been proposed by Lin et al. [19] and by Elias et al. [7] did not exploit UAV systems, it is worthwhile to consider them for their contributions to photogrammetric methods that can be applied for water level estimation. In particular, Lin et al. [19] have exploited images that were acquired by a surveillance camera, while Elias et al. [7] developed a procedure that is based on the use of smartphone images. Both of these methods process a series of images that have been acquired in a short period of time in order to reduce the noise effects by calculating the mean image. At this point, high frequency filters are applied to the mean image and line extraction algorithms, like the Hough transform, are implemented in order to retrieve the water surface line. Such algorithms, which have also been described by Isidoro et al. [20], have been used in this article too, as will be shown in the following sections. Lin et al. [19] have used local water gauges both for the calibration between the object space and the image space and for camera movement detection by least-squares matching and normalized cross-correlation. Thus, by exploiting all of the collected data, i.e., by combining computer vision techniques and photogrammetric principles, collinearity equations can be written in order to be able to estimate the water level. On the other hand, Elias et al. [7] have developed a specific algorithm by which the water line is detected through the use of line-extraction code, as mentioned before. In addition, the algorithm projects the previously recorded 3D data into a synthetic image (rendering) representing the same local situation as the real camera-derived image. At this point, the real and the synthetic images are matched by the use of feature-based methods in order to establish the 2D–3D correspondence that is necessary for a subsequent space resection. At the end of the process, the detected 2D water line is transferred into object space and thus into metrically scaled 3D water-level values. Furthermore, Elias et al. [7] have developed a smartphone-based method for quantifying flood water levels using image-based volunteered geographic information (VGI). Herein, digital image processing and a photogrammetric method were combined in order to determine water levels. In particular, the random forest classification was exploited in order to simplify the ambient complexity and the HT-Canny method was applied in order to detect the flooding line of the classified image-based VGI. Hence, by combining the photogrammetric method and a fine-resolution digital elevation model that is based on the UAV mapping technique, the detected flooding lines were employed in order to determine the water level. Moreover, Lin et et al. [21] have aimed at defining water level changes during different tidal phases using a digital surface model (DSM) that is captured by a UAV in conjunction with a global navigation satellite system (GNSS). Here, the authors applied the SfM method that is also described in [9,11,12,13,14] in order to reconstruct a 3D scene geometry from a set of images. Using the structure from motion (SfM) algorithm, a DSM and orthomosaics were produced. Moreover, the GNSS provided horizontal and vertical geo-referencing for both the DSM and orthomosaics during post-processing, after the field observation at the study area. Ridolfi et al. [22] developed a methodology implementing a sensing platform that is composed of a UAV and a camera in order to determine water levels. In the mentioned work, the so-acquired images were analyzed using the Canny method to detect the edges of the water level and of the ground control points (GCPs) that were used as reference points. The water level was then extracted from the images and compared to a benchmark value that was obtained by the use of a traditional device. Nevertheless, SfM methods have to face problems that are related to water refraction. In order to address this, some image-based refraction correction algorithms were developed [23], while other works focused on machine learning for the classification of dense point clouds for refraction correction [24,25]. Other problems affecting both bathymetry and water level estimation, as already mentioned in this section, are related to the noise or reflections within the image. A critical point affecting all of the methodologies is the need for references that, as previously described, can be of different types: field measurement, fluid dynamics equations, ground control points, etc.

In this paper, a specific method for the evaluation of water levels was developed by using photogrammetry from images that were collected by a UAV. In particular, a dense point cloud was retrieved and the random sample consensus (RANSAC) method was used in order to find the plane that better fits the river water surface plane. A reference point of a known altitude within the image was then exploited in order to compute the distance between it and the fitted plane in order to obtain the altitude of the free surface of the river. By the means of this methodology, this paper further aims at making a summary of the critical aspects and strong points affecting the application of photogrammetric techniques for river water level determination, starting from the ones that have already been highlighted by the state of the art, such as the influence of reflections that can cause false detections and geometrical distortions and random noise influencing the accuracy of the water level estimation. In fact, the literature highlights that the sources of uncertainty in water level computation methods, or in photogrammetry in general, are associated with image-based measurement systems (image focus and resolution, perspective, lens distortion, lighting effects, UAV flight altitude, yaw angle and viewing angle) [22,26,27] and connected to the physical and environmental characteristics of the study area (the meniscus that water forms at the point of contact with a background because of surface tension forces [26], the combination of waves and both the angle and intensity of the incoming light source, the presence of clouds and fog [22] and the vegetation density and terrain slope [28]). In this scenario, a great contribution is also given by the exploited algorithms themselves [28,29] and this is the reason why it is extremely important to characterize each new methodology in terms of the influences of the algorithmic parameters. This is valid not only for UAV photogrammetry applications but also for aerial light detection and ranging (LIDAR) techniques [30] that can be used for water level computation. For this purpose, in this paper an analysis of the sensitivity of the water level estimation algorithm that was used to process the parameters has been also performed.

In Section 2, the workflow for rivers’ water level evaluation is presented and each step of the workflow is described. In Section 3, the authors show the results and report a sensitivity study about the process parameters of the exploited algorithm. Within this same section, a discussion about the results and a comparison with LIDAR used for the same applications is provided, while in Section 4 the conclusions are outlined.

## 2. Materials and Methods

The workflow for determining the altitude of the free surface of the river from aerial images is shown in Figure 1 and can be described as follows.

### 2.1. Analysis of the Area under Study

Before conducting the flyover, it is necessary to visit the site in order to check for obstacles and to establish the shooting height and angle. The chosen body of water for this study was the Piave River, a tributary to the Centro Cadore Lake, which is an artificial lake in the province of Belluno (Italy), see Figure 2a. The choice fell on this river for several reasons. Firstly, the height of the free water surface of the river is related to the height of the upstream lake, which is artificially controlled. For this reason, it is possible to know in advance the height of the free water surface at any time of the day. Moreover, this river is particularly suitable for the test because it is a perfect real-life case study, not easily accessible and full of disturbing elements such as tall trees, pylons, bridges, rocks and bends. In addition, the lake undergoes lamination throughout the year and the water level can vary more than 10 m during the course of the year.

### 2.2. Mission Planning

Google Earth Pro was used to select the coordinates of the UAV mission. The Litchi software for DJI UAVs was used to establish the trajectory and shooting data [31]. Once the mission start point (A) and mission end point (B) had been defined, the number of images to be recorded and the shooting data were determined. The software automatically discretized the trajectory into a series of n points, from point A, i.e., p (0), to point B, i.e., p (n). The UAV moved to the first point and captured an image, then it moved from point to point until the end of the trajectory, at the position p (n).

### 2.3. Image Collection

The UAV was programmed to move from point A to point B (200 m of distance) at an altitude of 30 m from the ground level, for a total of 200 frames. The UAV’s trajectory from 46°27′20.11′′ N 12°25′5.01′′ E to 46°27′24.46′′ N 12°25′11.54′′ E is reported in Figure 2b. The coordinate position of each image, taken from the UAV’s on-board GPS system, was stored in the image metadata in order to help the photogrammetric software to align the photos properly. The UAV that was used for the image recording is the DJI Mavic 2 Pro (equipped with 20 MPx 1” RGB CMOS sensor and 35 mm f 2.8 lens with a 77° field of view—FOV). The pitch angle of the UAV’s camera was fixed at 45 °. An example of an image that was recorded by the UAV at 30 m with 77° FOV with a pitch angle of 45° is shown in Figure 3.

### 2.4. Point Cloud Computation

The software that was used for the point cloud computation is AGISOFT Metashape, a stand-alone software product that performs the photogrammetric processing of 2D images and generates 3D spatial data that can be used in several applications. The images that were recorded in the previous step were imported into the AGISOFT Metashape software. The images (some of which are reported in Figure 3) were firstly aligned, then the dense point cloud was computed with the parameters that are reported in Table 1. After this processing, a 3D point cloud was obtained as the output (Figure 4). The point coordinates were then exported in an ASCII text file with the local coordinates, in meters, in order for these data to be processed with external software.

### 2.5. River Water Surface Plane Extraction

The pyRANSAC-3D Python implementation of the random sample consensus (RANSAC) method was used in order to find the plane that best fits the river water surface plane. This algorithm is able to fit primitive shapes (line, cylinder, cuboids, and planes) into a point cloud. The plane primitive is chosen in order to find the water surface plane. The algorithm finds the best equation of the plane that maximizes the point cloud inliers, based on a threshold that randomly takes 3 different points of the selected point cloud for each iteration. For this purpose, the algorithm takes 3 parameters as its inputs. The first parameter is the *threshold*: the maximum distance from the plane that is required in order to consider a point to be an inlier. This threshold can be also seen as the depth of the bundle of planes fitting the river water surface. The second parameter is the *minPoints*: the minimum number of inlier points that a selected plane must have so as not to be discarded. The third parameter is the *maxIteration*: the number of maximum iterations which the algorithm will loop over. As its output, the algorithm then returns the plane equation β: ax+by+cz+d=0 and the array of points that are considered to be inliers. Figure 5 shows how the RANSAC algorithm automatically detects the water surface plane; the black dots represent the point cloud and the red dots represent the river water surface plane inliers.

### 2.6. Known Point Selection

After the determination of the equation of the river water surface plane, a reference with a known altitude is needed for the relative altitude computation. This reference must be time invariant. This feature can be a point or a more complex geometrical entity, such as a horizontal plane that is parallel to the water surface. This latter kind of feature provides lower uncertainty and a higher level of consistency over time but is not always available in the territory. Therefore, it was decided to use a reference point, due to its common presence in the study area and due to the fact that it is more conservative in terms of uncertainty. Hence, a reference point had to be manually selected from the point cloud. In this particular case it was easy to select a point belonging to the road over the bridge that is known to be positioned at 685 m AMSL.

### 2.7. Estimation of the Altitude of the Free Surface of the River

Knowing the positions $p_{x},p_{y},p_{z}$ of the previously selected point and the equation of the plane belonging to the water surface plane, the altitude (above sea level) of the river water’s free surface is easily calculated by subtracting the altitude of point p from the distance point-plane p and β (Figure 6, Equation (1)). This approach also works well in all of the cases wherein the resulting point cloud is randomly skewed because the distance that is calculated is always perpendicular to the target plane.
(1)$$h=p_{altitude}-\frac{\left|ap_{x}+bp_{y}+cp_{z}+d\right|}{\sqrt{a^{2}+b^{2}+c^{2}}}$$

## 3. Results

Ten flyovers were carried out on separate dates, over the course of a month, when the height of the water level was different. The method that is described above was applied to the ten obtained datasets of images and the results were compared with the data that were provided by the manager of the dam upstream of the river.

In order to simulate the use of image sensors with lower resolution and/or framing at higher distances, the method was applied to the same subsampled dataset in such a way that 20, 12, 6 and 2 megapixel-sized images were used as the input for the point cloud computation. Table 2 shows the results of the point cloud computation of the datasets with different resolutions. These results show how, as the resolution is reduced, the number of points that are detected by the photogrammetry software increases. This is probably because, by lowering the level of detail, METASHAPE’s image alignment algorithm can more easily find matches between the input images. This results in a more detailed point cloud, as evidenced by the increase in the total number of the points that were computed. By increasing the point cloud quality, the number of points that are associated with the river’s edge also increases, as is reported in the third column of Table 2. Losing resolution, in fact, compensates for all of the noise effects that are introduced by natural elements in the footage, such as the flow of the water or the movement of tree branches in the wind. For these reasons, the algorithm was tested in the 2 MP configuration.

The results of the 10 flyovers using images at 2 MP are reported in Figure 7. As far as the calculated water level values are concerned, it can be seen that the proposed method corresponds well to the reference values with an R2 = 0.98, which indicates a very strong correlation between the two approaches. In fact, the linear fit of the values of the water level shows a slope of 0.95 and a standard deviation of 0.37 m.

### 3.1. Uncertainty and Sensitivity Analysis

#### 3.1.1. Sensitivity to Image Reflection

It is interesting to note the possibility of having some artifacts that could affect the points cloud computation and, therefore, the water level estimate’s accuracy. In the case of a flat water surface and in particular lighting conditions, specular reflections may occur (as is evident in Figure 8) where the reflections of clouds, of the bridge and of other “objects” that are present on the banks are visible.

These reflections can be interpreted by the photogrammetry program as “virtual” objects under the water level, making it sometimes difficult to recognize the panorama.

This condition is not very frequent and its effects can be minimized with the appropriate choice of the UAV’s mission paths. Furthermore, although the point clouds are not of very good quality when they are viewed at glance, the algorithm that is proposed in this article is quite immune to this problem by focusing on a plane that contains a dense population of points, which does not necessarily define a half-space.

#### 3.1.2. Sensitivity to Algorithm Input Parameters

The parameters that are considered to have influenced the accuracy of the algorithm for the plane extraction process are:the threshold or depth of the bundle of planes fitting the river water surface;the maximum number of iterations.

The threshold, i.e., the plane bundle depth, has been varied between 0.05 m and 2 m, for a total of 24 values, while the number of iterations has been ranged from 1000 to 100,000, for a total of 5 values.

Furthermore, the sensitivity of the algorithm to noisy point clouds has been tested. In order to create noisy point clouds on purpose and to control the level of the noise, a Gaussian distribution was created with the same dimensions of the point cloud and it was then added to the point positions. Different levels of noise have been considered, from 0 (i.e., no noise added) up to 4.5 m, for a total of 10 values. Those values were used as the mean values of the Gaussian distributions that were created. This is intended to take into account:conditions in which the software would have difficulty in correctly reconstructing the point cloud, such as in the case of images that have been affected by thermal noise, low ambient light, dusk, dawn, long exposure times or the presence of haze, fog or rain;the repeatability and reproducibility of the process in reconstructing the point cloud from several images that were recorded at the same time.

The absolute error in the water level evaluation, with respect to the known ground level depth of 18 m, has been calculated for the 1200 different combinations of cases, i.e., 24 plane depth values, times 5 iteration numbers, times 10 levels of noise.

The maps of the absolute levels at different noise levels that were simulated on the point clouds are reported in Figure 9. It can be deduced that the optimal plane depth is 0.5 m since, from this value, the error level exhibits a drop from a maximum of 20 m to a mean value of 4.9 m. The error increases with the increase in the plane depth but it is not affected by the number of iterations. From the previous maps it is possible to extract slices at different number of iterations, for instance 1000, 10,000 and 100,000, showing the trend of the absolute error in the function of the plane depth, for the different noise levels. Those trends are illustrated in Figure 10 which show that, for a low number of iterations—see Figure 10a, the absolute error assumes important levels, except in the case of plane depths between 0.4 and 0.7 m. When the number of iterations increases, for instance up to 100,000—see Figure 10c, the absolute error decreases regardless of the noise level, even if the minimum error is always concentrated as it is in the cases of plane depths between 0.4 and 0.7 m.

If slices at different plane depths are extracted, as illustrated in Figure 11, it is possible to show that when a small threshold is set the error reaches 20, see Figure 11a. When the threshold is 0.5, the error presents its minimum value, as visible in the Figure 11b, and when the threshold increases the error increases again.

For a better explanation, Figure 12 shows the superimposition between the point cloud and the resulting plane, in two extreme cases:Threshold = 0.05 m—Figure 12a, giving as an output a revealed plane, very thin and sloping, that is significantly different from the actual one;Threshold = 2.00 m—Figure 12b, resulting in a correct plane, but one which is too thick, giving a consequent error.

### 3.2. Discussion

In this subsection, the analysis of the uncertainty of the measurement system and of its sources is discussed. These sources are related to the random noise that can affect the point clouds and to the parameters that must be set in the algorithm for the estimation of the water plane (plane depth and number of iterations). By observing the uncertainty analysis results, it has been deduced that the source of uncertainty to which the algorithm is most sensitive is the plane depth, which can give an important bias (up to 20 m) if it is set to extremely low values (below 0.05 m), as evidenced in Figure 9 and Figure 10. In addition, with regard to the number of iterations, it has been observed that when the number increases, the absolute error decreases regardless of the noise level, even if the minimum error is always concentrated as it is in the cases of plane depths between 0.4 and 0.7 m; see Figure 11. Therefore, when the plane depth assumes a reasonable value, e.g., between 0.4 and 0.7 m, the uncertainty that is related to the other sources (noise and number of iterations) is kept to limited levels (below 2 m). In this case, the effects of noise and of the number of iterations do not affect the uncertainty.

Thus, by the exploitation of the optimal parameters, the comparison with the data that were provided by the manager of the dam upstream of the river shows an accuracy level of around 100%. In fact, the linear fit of the values of the water level shows a slope of 0.95 and a standard deviation of 0.37 m; see Figure 7. Similarly, accurate results can also be obtained by LIDAR-based techniques, as shown in Paul [31], wherein a relative error of 0.1% is declared in spite of the low reflectivity of the air–water interface to near infrared radiation (NIR). However, LiDAR methods are expensive and extensive field data collection can be an extremely challenging task for large river studies [11].

## 4. Conclusions

This paper presents a methodology that is based on photogrammetry and may be used to measure the altitude of river water without the need for a reference artifact. The method uses images that, in the specific case that has been discussed, have been acquired by an RGB camera that was mounted on a UAV system in order to reconstruct dense point clouds. From the point cloud, the river water surface plane was retrieved by applying the random sample consensus (RANSAC) method.

The procedure has been applied for the retrieval of the water altitude of a river that was monitored during three missions and the results, in all of the missions, were very promising. This demonstrates that the automatic identification of the altitude is robust and efficient.

The paper has also focused on the analysis of the uncertainty of the measurement system and of its sources, aiming at finding the optimal process parameters. In this regard, the optimal plane depth is 0.5 m and, in proximity to this value, the number of iterations has no influence on the error amount.

The proposed methodology is suitable for the monitoring of ungauged rivers. When the placement of continuous real-time meters is not feasible due to the orography or the river’s path, a survey by means of a UAV may represent a valid alternative. Due to the intrinsic discrete frequency of the data acquisition, the use of a UAV should obviously be contextualized to those cases which do not require a high data acquisition frequency (e.g., sub-daily). In addition, the uncertainty that has been described above is another factor to be considered for the actual applications of this methodology. Some contexts, for example the monitoring of urban drainage canals, require real-time and high accuracy measures, since in those scenarios just a small increase in the level may result in a flood, and monitoring by a UAV is not a proper solution. Conversely, survey campaigns that are aimed at studying the overall river’s behavior during an extended period could benefit from the use of UAV measures. Specific objectives could be the study of rainfall-induced effects on rivers when the peak flow duration spans over several days or, conversely, the effects of drought or other sources of flow reduction from irrigation demands. Moreover, the estimation of the environmental flow and similar measurement campaigns that have aimed at defining strategic plans represent a suitable context for the application of UAV photogrammetry.

## Figures and Tables

**Figure 1 sensors-22-05319-f001:**
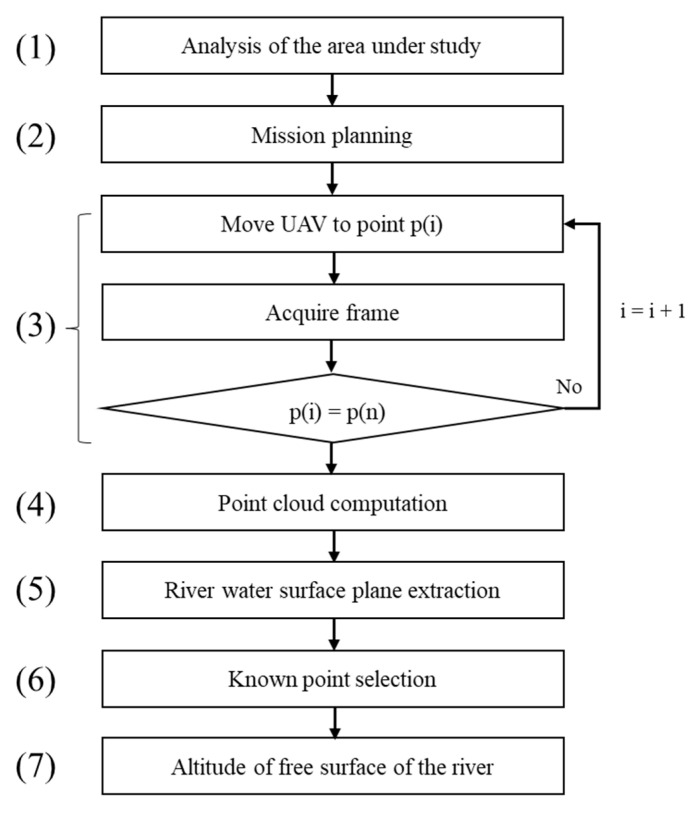
Workflow for determining altitude of the free surface of the river for aerial images.

**Figure 2 sensors-22-05319-f002:**
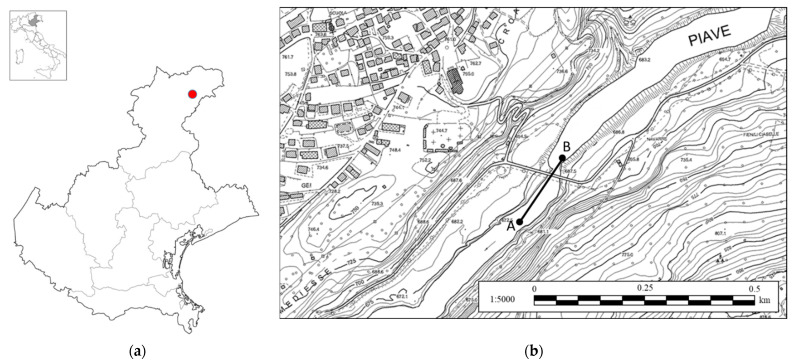
Mission planning: (**a**) geolocation of the study area; (**b**) mission designed on Google Earth in order to move the UAV from point A to point B.

**Figure 3 sensors-22-05319-f003:**
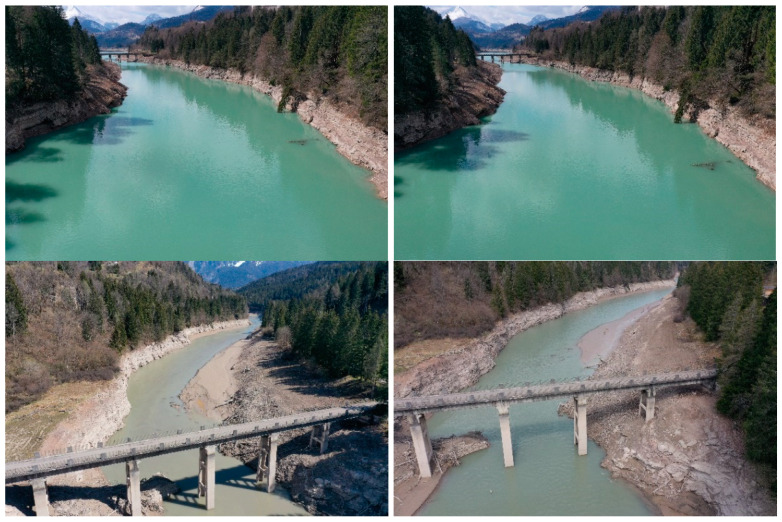
Example of images recorded by the UAV with FOV = 77° and gimbal pitch angle = 45°.

**Figure 4 sensors-22-05319-f004:**
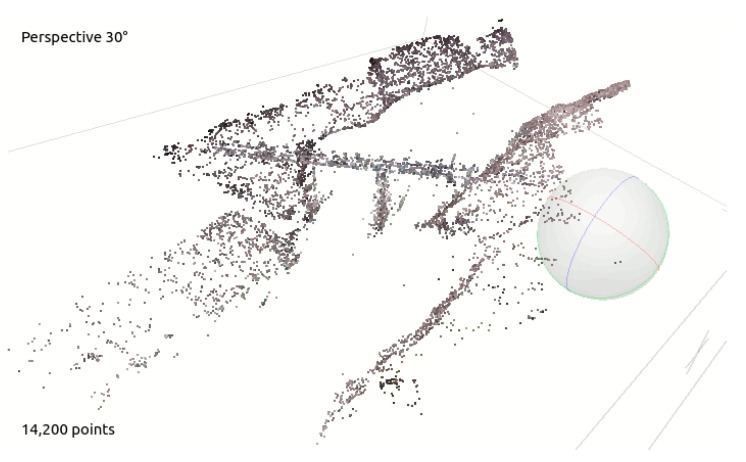
Point cloud obtained from river images collected by the UAV.

**Figure 5 sensors-22-05319-f005:**
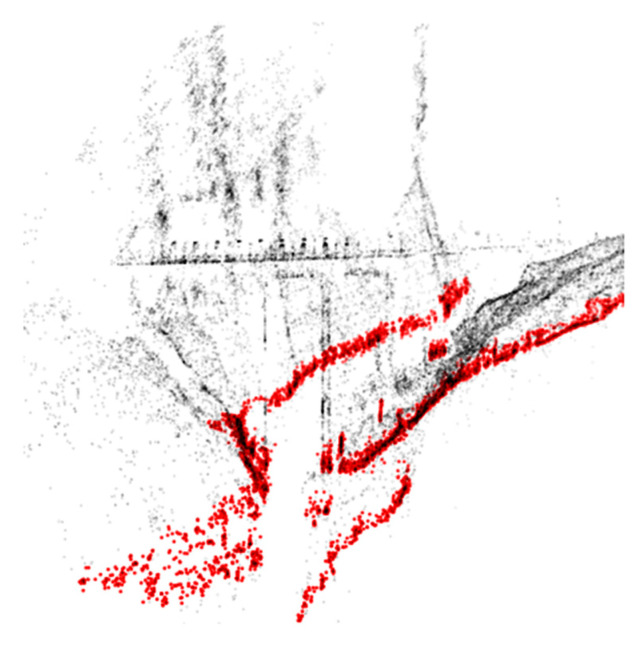
Black dots = point cloud. Red dots = river water surface plane inlier points.

**Figure 6 sensors-22-05319-f006:**
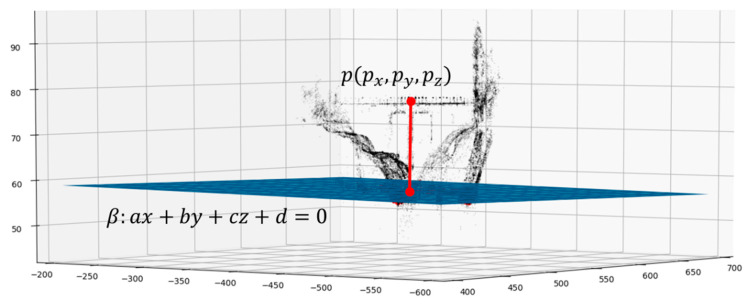
Distance between the plane of the water surface and the point of known altitude.

**Figure 7 sensors-22-05319-f007:**
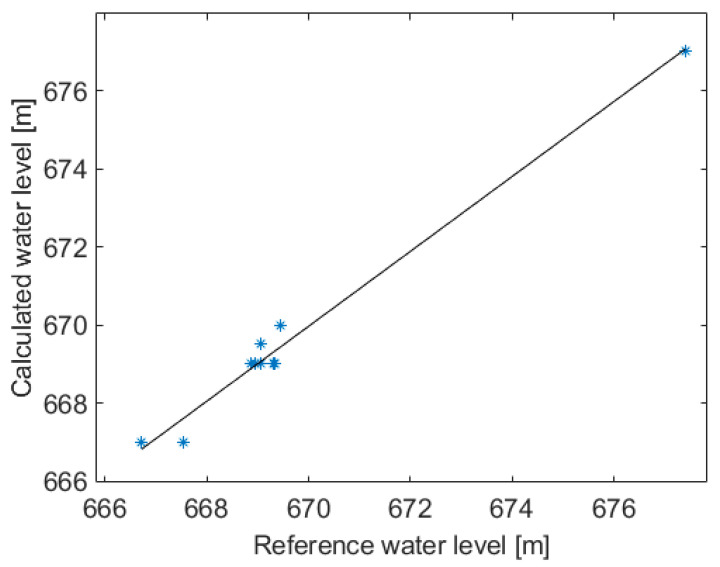
Correlation between calculated water level and reference water level.

**Figure 8 sensors-22-05319-f008:**
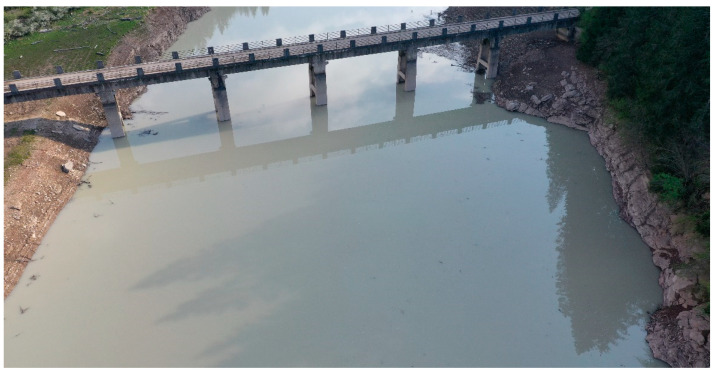
Specular reflection of bank vegetation and the bridge on the flat water.

**Figure 9 sensors-22-05319-f009:**
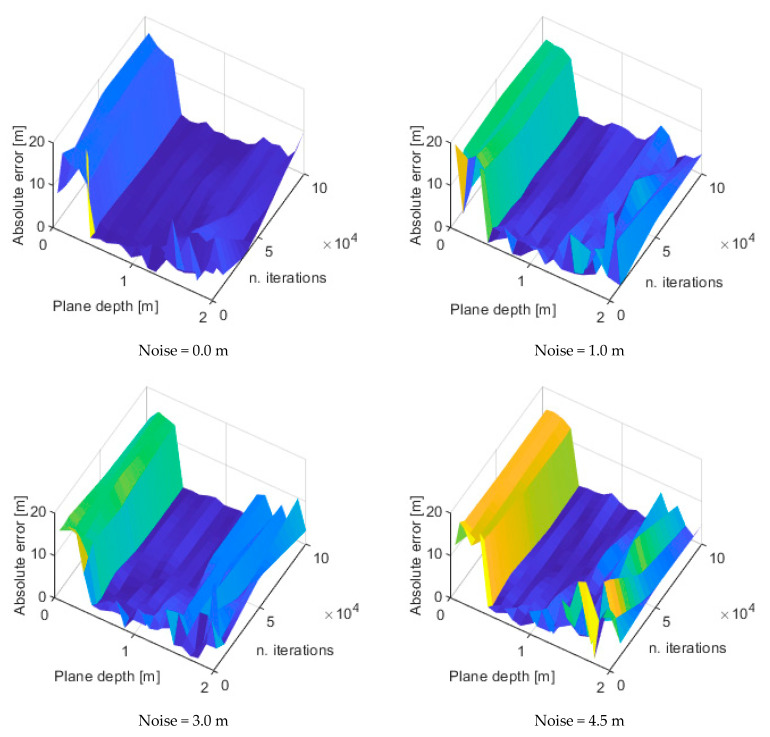
Error maps for different noise levels.

**Figure 10 sensors-22-05319-f010:**
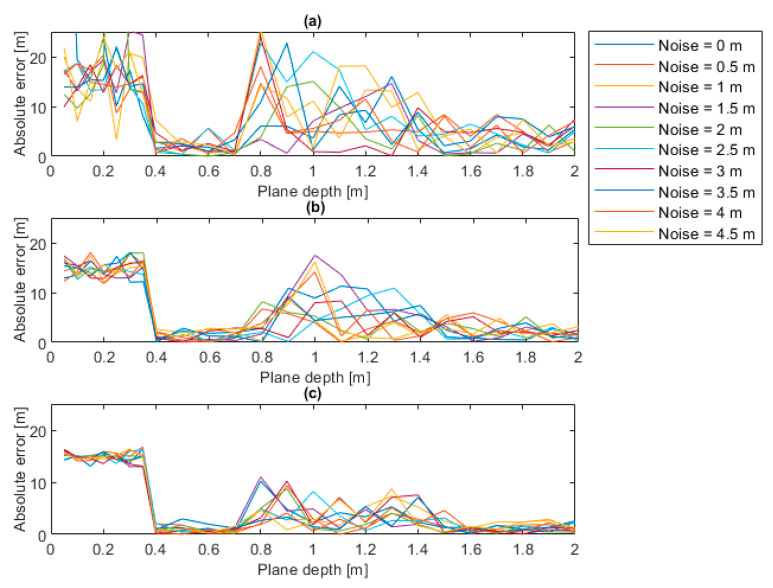
Error trends for different noise levels (**a**) 1,000 iterations, (**b**) 10,000 iterations, (**c**) 100,000 iterations.

**Figure 11 sensors-22-05319-f011:**
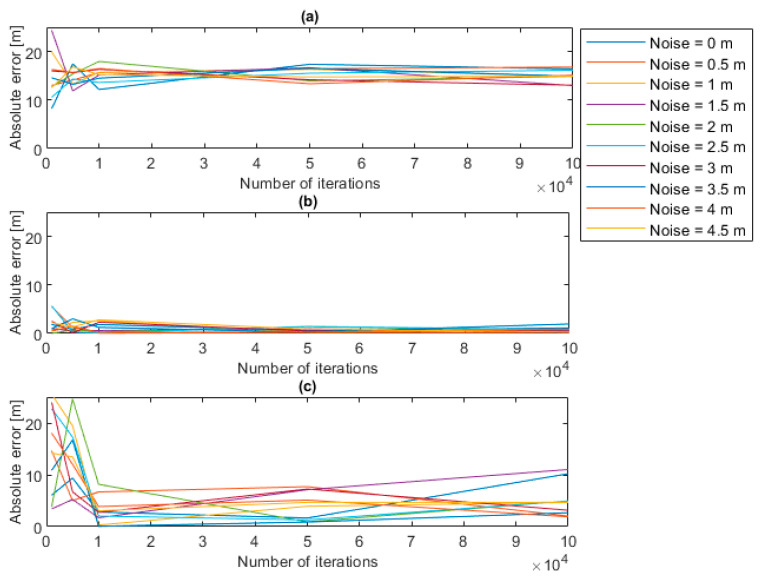
Error trends for different noise levels (**a**) plane depth of 0.05 m, (**b**) plane depth of 0.5 m, (**c**) plane depth of 2 m.

**Figure 12 sensors-22-05319-f012:**
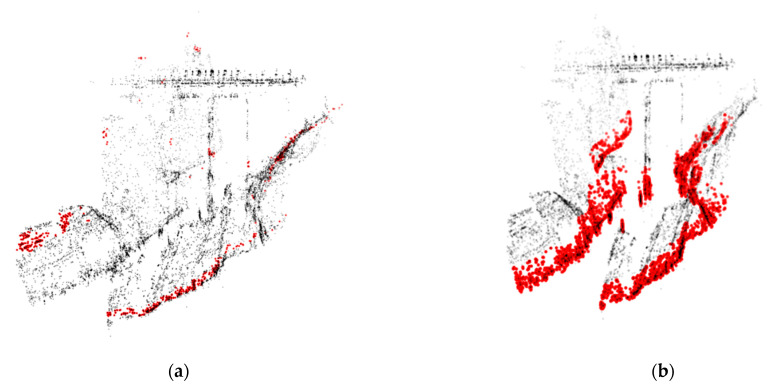
Plane resulting from two different thresholds (0.05 m (**a**) and 2.00 m (**b**)).

**Table 1 sensors-22-05319-t001:** Parameters used for point cloud computation. References can be found in the documentation of the software Metashape [32].

Photos Alignment	
Accuracy	Highest
Key point limit	40,000
Tie point limit	4000
Guided image matching	OFF
Adaptive camera model fitting	ON
**Dense Cloud Computation**	
Quality	Ultra High
Depth filtering	Mild

**Table 2 sensors-22-05319-t002:** The number of detected points within the point cloud and the points associated with the river’s edge in relation to the image resolution.

Resolution [MP]	Total Points	River Edge Points
20	19,656	4012
12	22,600	4265
6	36,433	8552
2	53,001	12,722

## Data Availability

The raw data were generated at Università Politecnica delle Marche. The derived data supporting the findings of this study are available from the corresponding author on request.

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
