# Peer review of "Rivers’ Water Level Assessment Using UAV Photogrammetry and RANSAC Method and the Analysis of Sensitivity to Uncertainty Sources"

_sensors, 2022, doi:10.3390/s22145319_

Round 1

Reviewer 1 Report

1. Workflow is too simple

2. Figure 2:irregular map

3. Impact analysis is too simple

4. Need more references

Author Response

  1. Workflow is too simple
    Thank you for your comment. Work flow has been detailed, providing information about the analysis of the study area and specifying the for cycle used for image collection.
  2. Figure 2:irregular map

The figure has been modified including the map of the river site provided with a geometrical scale (Figure 2b). The map for geolocation has been also added as Figure 2a.

  1. Impact analysis is too simple

Thank you for the comment. A more detailed explanation about the work impact has been added at the end of the introduction, in order to highlight the importance of performing a sensitivity study on all the image-based and photogrammetry-based techniques, because of the various uncertainty sources associated (light conditions, reflections, environmental conditions, parameters used within the exploited algorithms). A Section dedicated to the discussion about this sensitivity study has been also added, and a comparison with LIDAR-based techniques is reported.

  1. Need more references

Additional references have been added within the manuscript for a better explanation of the work impact, as mentioned in the previous comment.

Reviewer 2 Report

The manuscript presents a study which makes use of high spatial resolution of UAV-based imagery to determine river water level, RANSAC (Random sample consensus) was used for this purpose.

The manuscript addresses an interesting topic but several changes and explanations are required before it could be considered for acceptance.

Comments and suggestions:

Missing discussion, authors could use some of the content from the studies presented in the Introduction to create the discussion and, at same time, reducing the introduction. Some content from Conclusion can also be used.

When citing a study after using "in" author names should be mentioned (e.g. In Legleiter [10] ...").

Line 142: "for" or "from"?

Section 2 can be divided into several subsections, one for each step of the workflow. Moreover, a subsection describing the study area should also be added, my suggestion is that subsection should be the first (2.1.), and a figure showing its location should be added.

Add a scale bar to Figure 2.

Consider avoiding the use of "drone", instead use UAV.

In the abstract the authors state "from videos recorded by UAV." while in Section 2 (paragraph in line 146) it is stated that the UAV "taking a picture every 2 seconds for a total of 200 pictures", please clarify if a video was recorded or photos. Moreover, state the software that was used for mission planning or to pilot the UAV.

Line 160; "point cloud computation" instead of "point cloud estimation".

Please specify the parameters used in Agisoft Metashape specially the point density used for point cloud computation. As it seems that few points were generated (Figure 4).

Please mention type of data and its benefits and limitations that could also be used such as LIDAR.

Author Response

Missing discussion, authors could use some of the content from the studies presented in the Introduction to create the discussion and, at same time, reducing the introduction. Some content from Conclusion can also be used.

Thank you very much for your suggestion. A section related to the discussion of results has been added, in order to highlight the importance of performing a sensitivity study in terms of the used process parameters, in order to find the optimal ones and to obtain high accuracy results.

When citing a study after using "in" author names should be mentioned (e.g. In Legleiter [10] ...").

Thank you. The author’s name has been included in the text when needed as suggested by the reviewer.

Line 142: "for" or "from"?

It is “from”, thank you for the correction.

Section 2 can be divided into several subsections, one for each step of the workflow. Moreover, a subsection describing the study area should also be added, my suggestion is that subsection should be the first (2.1.), and a figure showing its location should be added.

Thank you for the suggestion. The subsections have been created, including the first one related to the analysis of the study area.

Add a scale bar to Figure 2.

The image in Figure 2 has been replaced by a more detailed map including a scale bar.

Consider avoiding the use of "drone", instead use UAV.

Thank you, the correction has been made.

In the abstract the authors state "from videos recorded by UAV." while in Section 2 (paragraph in line 146) it is stated that the UAV "taking a picture every 2 seconds for a total of 200 pictures", please clarify if a video was recorded or photos. Moreover, state the software that was used for mission planning or to pilot the UAV.

Thank you. The word “videos” in the abstract has been replaced by “images”, since the UAV is able to record only images.  The software used for mission planning is Litchi. It has been specified within Subsection 2.2, and the website reference has been added.

Line 160; "point cloud computation" instead of "point cloud estimation".

Thank you. The correction has been made.

Please specify the parameters used in Agisoft Metashape specially the point density used for point cloud computation. As it seems that few points were generated (Figure 4).

A table, specifically Table 1, has been added in the manuscript reporting the parameters used in Agisoft Metashape. The reference to the software documentation has been added for further information.

Please mention type of data and its benefits and limitations that could also be used such as LIDAR.

Thank you for the comment. References to LIDAR have been included within the Introduction, and in the Discussion Subsection a brief comparison with the methodology developed in this work has been considered.

Reviewer 3 Report

I have no significant comments. I perceive the main goal of the article as the authors' efforts to expand knowledge in UAV technology in combination with the developed specific method of water level assessment using photogrammetry from videos recorded by UAV and also by analyzing the sensitivity of the water level estimation algorithm to process parameters. These skills may be of interest to SENSORS readers.

Special commnets: I recommend adding points A and B in Figure 2, which identify the movement of the UAVs during the experiment. In lines 318-320: Acknowledgments, it is, in my opinion, irrelevant.

Summary: I recommend the paper, after minor corrections, for publication within an academic and research discussion on an issue that contributes to the expansion of current knowledge.

Author Response

I recommend adding points A and B in Figure 2, which identify the movement of the UAVs during the experiment.

We thank  the reviewer for the comment. The points A and B have been added in the figure, that has been detailed with a more specific map including the geometrical scale.

In lines 318-320: Acknowledgments, it is, in my opinion, irrelevant.

It was a mistake leaving the default sentence. Thanks for having pointed out this problem.

Reviewer 4 Report

In this paper authors give an analysis of a system for river water level determination from images taken with a drone and using the RANSAC method. Authors used standard software and libraries for implementation.

Remarks on the paper:
Abbreviation UAV in abstract should be explained.
Some results should be pointed out in the abstract.
Abbreviation SfM in line 112 should be explained, "f" should be capital or small letter in 113 its capital.
In line 154 add FOV after Field of View.
Authors added synthetic noise to images using gaussian distribution. Examples should be given in a real noisy environment for example with the presence of fog. In low light situations, dusk, dawn, rain falling, etc.
Definitely more tests should be carried out for different water levels. Only 3 tests are shown with only two different water levels.
How does the choice of the reference point affect the accuracy of the method. Why not use another plane for the reference point. Based on the user selected point and other adjacent points a reference plane is calculated.
Authors used 20 MP camera, what if the resolution is lower or higher, tests should be conducted and results shown. Additionally, how does the height of the drone effect the results.

Author Response

Abbreviation UAV in abstract should be explained.

Abbreviation has been explained. Thanks for reporting the mistake.

Some results should be pointed out in the abstract.

The abstract has been provided with a synthetic description about the performed sensitivity study and its main results in terms of algorithm optimal parameters (plane depth and number of iterations).

Abbreviation SfM in line 112 should be explained, "f" should be capital or small letter in 113 its capital.

Thank you for the comment. We made consistent the SfM acronym through all the text of the paper.

In line 154 add FOV after Field of View.

FOV has been added as suggested.

Authors added synthetic noise to images using gaussian distribution. Examples should be given in a real noisy environment for example with the presence of fog. In low light situations, dusk, dawn, rain falling, etc.

Actually, we have added gaussian noise to the computed point clouds and not to the input images as it is explained in line 339. This strategy has been adopted because we aimed to take into account any kind of haze effect.  A sentence to clarify this choice has been included from line 341 to line 346.

Definitely more tests should be carried out for different water levels. Only 3 tests are shown with only two different water levels.

A more consistent number of data has been considered for analysis, corresponding to ten flyovers carried out on separate dates, over the course of a month, when the height of the water level was different. The described method was applied to the ten obtained dataset of images and results were compared with the data provided by the manager of the dam upstream of the river. This analysis is shown in Section 3.

How does the choice of the reference point affect the accuracy of the method. Why not use another plane for the reference point. Based on the user selected point and other adjacent points a reference plane is calculated.

We have added a paragraph in Subsection 2.6 to describe the choice of a point instead of a plane.

Authors used 20 MP camera, what if the resolution is lower or higher, tests should be conducted and results shown. Additionally, how does the height of the drone effect the results.

The study has been carried out with a specific UAV, that is the one supplied in the field of the project in which the activity described in this work is included.

However, we thank so much the reviewer for the relevant comment, that enabled us to find an interesting result, i.e. the fact that as the camera resolution decreases, the number of detected points increases. This is probably because by losing the detail level, METASHAPE's image alignment algorithm can more easily find matches between input images. This results in a more detailed point cloud. Losing resolution, in fact, compensates for all noise effects introduced by natural elements in the footage, such as the flow of water or the movement of tree branches in the wind. For this reason, the whole analysis has been conducted in the worst condition (2Mpx) that is, however, the most efficient one. We added a paragraph at the beginning of Section 3.

Round 2

Reviewer 2 Report

The authors improved the manuscript and addressed the comments and suggestions from the previous review. 

However, there are still some issues related with the citations. It should be "LastName et al. [#]" and not "[#] LastName" or "LastName et al. in [#]". References should be numbered in the order they are first cited in the text. Also the two new added references no. 32 and 33 are not completely correct, as only a URL is provided.

Figure 2 can be improved, namely the size of Figure 2a, the authors can have both sub-figures side by side.

Author Response

Thanks to the reviewer for the comments.

Issues related to citations have been fixed, and the order of references has been checked.

References 32 and 33 have been detailed by addition of software specifications.

Figure 2 has been modified as suggested.

Reviewer 4 Report

Authors corrected allmost all remarks on the paper. Manuscript is good for acceptance.

Author Response

Thanks to the reviewer for the feedback.